## RESEARCH ARTICLE

# Interactions of husbandry, landscape, and immunity in regulating viral loads for managed honey bees

Allison Malay*, Rachel Weavers and Kenneth M. Fedorka

## ABSTRACT

The western honey bee, *Apis mellifera*, continues to experience widespread die-offs that threaten their critical ecological and agricultural roles. Given the recognized impact of viruses on the increased mortality rates, it is imperative to understand the forces shaping viral infections. In this study, we explore how hive husbandry, landscape, and immunity influence viral loads in managed bees. We characterized 43 apiaries across Central Florida for eight husbandry interventions, five landscape variables, transcription of four immune genes, and infection intensities of four viruses: Black Queen cell virus (BQCV), deformed wing virus type A (DWV-A), Lake Sinai virus (LSV-2), and Israeli acute paralysis virus (IAPV). We found that colonies surrounded by more floral resources and fresh water bodies were associated with increased viral loads and increased viral coinfections. We speculate that increased floral resources increased pollinator abundance, thereby increasing transmission rates and viral richness. We further speculate that increased open water similarly increased pollinator abundance and/or exposure to immunity-altering pesticides. Last, we show that husbandry interventions aimed at reducing *Varroa destructor* mites can have positive and negative off-target viral impacts. Our data underscore the importance of landscape, immunity, and husbandry in honey bee disease dynamics and highlight the complexity of their interactions.

KEY WORDS: Amplification effect, Bee forage area, GIS, Immune gene transcription, Infection intensity

## INTRODUCTION

Since its introduction into North America, the western honey bee (*Apis mellifera*) has become a critical component of agricultural productivity and ecosystem health. Within the United States, honey bees have an estimated 17.4-billion-dollar impact on crop production, not including their honey and beeswax market contributions (Calderone, 2012). They are essential pollinators for at least 130 types of fruits and vegetables that are key to a healthy human diet (Degrandi-Hoffman et al., 2019). Honey bees have also displaced numerous native bees in natural habitats (Page and Williams, 2023) and have become the most common floral visitor worldwide (Hung et al., 2018) making them significant components of pollinator communities and integral to ecosystem stability (Papa et al., 2022). However, managed honey bee colonies have in recent years exhibited significantly elevated mortality rates, with US beekeepers reporting average annual losses of 30-50% between 2011-2022 (Aurell et al., 2024). This makes understanding the factors that influence honey bee health an important priority to ensure the continuation of agricultural and ecosystem services.

A key factor in honey bee health is parasitic infection, including macro- and microparasites. The macroparasitic small hive beetle (SHB), *Aethina tumida*, can induce colony stress by damaging comb, consuming brood and honey stores, and promoting hive abandonment (Neumann and Elzen, 2004). The macroparasitic mite *Varroa destructor* can weaken bee immune defense by feeding on fat body (an important immunological structure in insects) while altering immunological proteins and hemocyte abundance (Posada-Florez et al., 2019; Richards et al., 2011). *V. destructor* is also a major vector of the notorious microparasite deformed wing virus (DWV), which is strongly associated with overwinter colony loss (Natsopoulou et al., 2017). In addition to insect vectoring, DWV is transmitted through oral–oral (trophallaxis) and oral–fecal pathways (Gisder et al., 2009; Martin and Brettell, 2019; Vilarem et al., 2021), which adds to its persistence within a colony. DWV currently infects approximately 55% of colonies worldwide and 95% of US colonies (Martin and Brettell, 2019). Several other viruses also pose significant threats to honey bees, such as Black Queen cell virus (BQCV), Lake Sinai virus (LSV), and Israeli acute paralysis virus (IAPV; Gisder and Genersch, 2017). These viruses can infect bees simultaneously and impart a heavy toll on colony health, as colonies with a greater number of co-infections are more likely to fail (vanEngelsdorp et al., 2009).

The landscape in which colonies exist also plays a key role in their health. Human encroachment into natural areas often reduces the availability and diversity of floral resources that can impact bee nutrition, stress, and disease incidence (DeGrandi-Hoffman et al., 2019; Chen et al., 2021; Tauber, 2020; Proesmans et al., 2021). The way colonies are impacted depends on the type of encroaching landscape. For example, colonies around agricultural development tend to have higher burdens of Varroa mites (Alburaki et al., 2018) and greater exposure to pesticides that induce lethal and sublethal effects (Boncristiani et al., 2012; Pohorecka et al., 2017; Traynor et al., 2016). Similarly, non-agricultural urban development can drive higher viral infection rates (Youngsteadt et al., 2015) and colony loss (Naug, 2009). Interestingly, some urban environments can provide more nutritional diets than natural lands due to higher horticultural turnover (Lecocq et al., 2015), which could improve colony health under certain circumstances. Thus, landscape structure can have profound but complicated effects.

With ongoing elevated annual losses, there is economic and ecological pressure to intervene. Accordingly, beekeepers have adopted a variety of mediations to improve colony success, such as removing macroparasites like small hive beetles and *Varroa* mites,

Department of Biology, University of Central Florida, Biological Sciences Building, 4110 Libra Dr., Orlando FL 32816, USA.

*Author for correspondence (al395458@ucf.edu)

A.M., 0009-0009-6426-6127; R.W., 0009-0009-9807-2558; K.M.F., 0000-0002-5514-8673

Biology Open

as well as providing additional nutritional resources in resource poor landscapes (Jack and Ellis, 2021). To reduce *Varroa* mite incidence, most beekeepers utilize chemical miticides (e.g. amitraz), though non-chemical mite interventions also occur (e.g. drone brood removal). While chemical compounds are effective at reducing mite loads, they can negatively impact colony health and productivity (Glavan, 2020; Qadir et al., 2021). Miticides have been shown to reduce food transfer among workers (Bevk et al., 2012), lower queen body size and reproductive capacity (Haarmann et al., 2002), and alter bee immune defense (Boncristiani et al., 2012; Glavinic et al., 2022; Jovanovic et al., 2022). How miticides impact bee viral loads, however, is still unclear (Boncristiani et al., 2012; O'Neal et al., 2017). Moreover, how landscape, husbandry, and immune function interact to impact the prevalence and intensity of viral infections is relatively unknown.

Here, we take an exploratory approach to characterize honey bee viral infections in relation to different husbandry, immunological, and landscape variables. Specifically, we surveyed Central Florida beekeepers on their husbandry treatments, characterized their apiaries' surrounding landscape via GIS, and sampled their colonies for bee immune gene regulation and infection with common viruses. We then used Generalized Linear Mixed Models (GLMMs) to explore how these variables associate with viral infection patterns. Honey bees live in a multivariate world and such information is critical to understanding the forces that shape honey bee disease.

## RESULTS
### Summary of surveys, sites, and viral loads
Of the 43 beekeepers sampled, most viewed themselves as hobbyists (81.3%), followed by sideliners (11.6%) and commercial beekeepers (2.3%). Most used miticides (72.1%) but few used non-chemical drone brood removal (9.3%). The most common miticides used were Apiguard and oxalic acid (34.9% and 27.9% respectively). Beekeepers self-reported an average of 21.43% colony loss, with a range of 0-62%. Landscapes were highly variable across the sites, with some sites in heavily urbanization areas (52% impervious surfaces (IS)) and others in more rural areas (1% IS). Among the sites, bee forage area (BFA) ranged from 0 to 33% (average 9%), natural lands (NL) ranged from 1 to 65% (average 25%), and agricultural lands (Ag) ranged from 0 to 48% (average 7%), while open water (OW) ranged from 1 to 39% (average 6.6%). For correlations among landscape variables and among immunity variables, see Table S3a and S3b, respectively. Of the 244 samples with detectable virus, 243 (99.6%) were positive for BQCV, 239 (97.9%) were positive for DWV, 109 (44.7%) were positive for IAPV, and 119 (48.8%) were positive for LSV.

### Impact of husbandry, landscape, and immunity
#### DWV
Of the husbandry variables examined, Apiguard, amitraz, and essential oils had significant associations with DWV infection intensity (Fig. 2; Table 1A). This makes sense considering these interventions are expected to reduce the abundance of DWV vectoring mites. However, these chemicals appeared to have opposing impacts, with Apiguard associated with reduced viral loads while amitraz and EO were associated with increased viral loads. Note that "essential oils" represents several products and does not differentiate between them. BFA had a positive impact on DWV, indicating that a greater abundance of pollination resources was associated with a greater amount of the virus. Both *domeless* and *dicer* were important predictors in the model; *domeless* was strongly negatively associated with DWV loads, whereas *dicer* showed a positive association.

**Table 1. Lowest AICc models for infection intensity of each virus as well as the first principal component of the combined viruses (PC Virus)**

| | Source | Beta | s.e. | SW | t-value | P |
|---|---|---|---|---|---|---|
| (A) | DWV Infection Intensity | | | | | |
| | *dicer* | 0.31 | 0.16 | 0.84 | 2.1 | **0.0444** |
| | *domeless* | −0.3 | 0.16 | 0.47 | −1.7 | 0.0937 |
| | Bee Forage Area | 0.23 | 0.13 | 0.53 | 1.9 | 0.0717 |
| | Amitraz | 0.3 | 0.13 | 1.00 | 2.4 | **0.0231** |
| | Apiguard | −0.4 | 0.13 | 1.00 | −3.4 | **0.0017** |
| | Essential Oils | 0.34 | 0.13 | 1.00 | 2.8 | **0.0078** |
| (B) | BQCV Infection Intensity | | | | | |
| | *ppo* | −0.3 | 0.15 | 0.51 | −1.8 | 0.0742 |
| | Open Water | 0.25 | 0.15 | 0.90 | 1.7 | 0.0969 |
| | Formic Acid | 0.33 | 0.15 | 1.00 | 2.3 | **0.0255** |
| (C) | IAPV Infection Intensity | | | | | |
| | *domeless* | −0.3 | 0.12 | 1.00 | −2.6 | **0.0136** |
| | Agricultural Area | −0.3 | 0.12 | 0.94 | −2.1 | **0.0426** |
| | Bee Forage Area | 0.55 | 0.12 | 1.00 | 4.7 | **0.0000** |
| | Open Water | 0.46 | 0.11 | 1.00 | 4.2 | **0.0002** |
| (D) | LSV Infection Intensity | | | | | |
| | Natural Lands | 0.32 | 0.15 | 0.74 | 2.2 | **0.0312** |
| | Open Water | 0.3 | 0.15 | 0.93 | 2.1 | **0.0396** |
| (E) | PC Virus Infection Intensity | | | | | |
| | *domeless* | −0.3 | 0.12 | 0.98 | −2.7 | **0.0112** |
| | Agricultural Area | −0.2 | 0.13 | 0.37 | −1.4 | 0.1761 |
| | Natural Lands | 0.51 | 0.13 | 0.58 | 4.0 | **0.0004** |
| | Open Water | 0.36 | 0.12 | 1.00 | 3.1 | **0.0043** |
| | Amitraz | 0.27 | 0.12 | 0.82 | 2.3 | **0.0290** |
| | Apiguard | −0.2 | 0.12 | 0.75 | −1.6 | 0.1123 |
| | Essential Oils | 0.22 | 0.12 | 0.57 | 1.9 | 0.0725 |

Presented models were distilled from global models that included all possible husbandry, landscape and immunity variables. β=standardized regression coefficient. SW=sum of Akaike weights. Bolded *P*-values and SW indicate important model predictors.

#### BQCV
Although formic acid is used to decrease mite abundance, our model suggested an off-target and positive impact of FA on BQCV infection intensity (Table 1B; Fig. 2). Likewise, the amount of open water surrounding the colony had a positive impact on BQCV. As *ppo* transcription levels increased, BQCV levels decreased suggesting a robust immune system helps to constrain viral infection levels.

#### IAPV
OW and BFA had significant positive impacts on IAPV infection intensity, while the proportion of Ag had a negative impact (Table 1C; Fig. 2). A negative association was observed between IAPV levels and *domeless* transcription. No husbandry variables emerged as significant predictors, suggesting IAPV is largely constrained by landscape and immune conditions.

#### LSV
Open water and natural lands landscape variables had positive impacts on LSV infection intensity, suggesting landscapes with more water and pollinator resources promoted a greater level of infection (Table 1D; Fig. 2). No immune or husbandry variables appeared to influence LSV intensity.

#### PC Virus
The first principal component of the four viral infection intensities represents an "overall" level of infection. It exhibited relatively equal positive loadings among the four viruses and accounted for 44.7% of variation. The immune gene *domeless* was a negative predictor of PC-Virus load. Landscape variables Natural

lands and open water were both significant positive predictors, while agriculture was a negative but insignificant predictor. Amitraz and essential oils were positive predictors of viral infection intensity.

## Viral richness, landscape, and immunity

The average number of viruses found in a colony was positively associated with bee forage area and open water and negatively associated with *domeless* (Table 2; Fig. 3A). To more fully examine the relationship between viral richness and immune variables only, we used a generalized linear mixed model using all available samples and including colony as a random factor. The number of pathogens infecting a bee sample was significantly associated with the transcription levels of the immune genes including *prophenoloxidase* (β±s.e. −0.48±0.08, $P<0.0001$; Fig. 3B), *defensin* (β=−0.17, $P=0.0142$), and *dicer* (β=0.27, $P<0.0002$).

## DISCUSSION

Honey bee colonies are experiencing substantial annual die-offs worldwide, which jeopardize their agricultural and ecological services. Given the detrimental consequences of these losses and the acknowledged role played by parasites, it is imperative to understand the forces that shape viral infections. Here we explored how husbandry, landscape and immunity variables interact to influence viral infection intensity. We found that all three variable types differentially impacted viral proliferation amongst four common viruses found in managed honeybee hives: DWV-A, BQCV, IAPV, and LSV-2. The viral-specific models generated herein highlight the complexity of these interactions and add important perspective to honey bee disease dynamics. To our knowledge this is the first study to assess the simultaneous impact of these variables on viral loads.

Regarding landscape, Bee Forage Area (BFA), Natural lands, and Open Water were the most consistently associated with viral infection intensity. BFA represents the amount of space surrounding a colony that contains floral resources. 'Natural lands' was highly correlated with BFA and represented the amount of natural resources surrounding a hive (similar to BFA). Variation in colony resource

availability can have profound effects on infectious disease dynamics (Becker and Hall, 2014). Theoretically, improved resources can decrease pathogen prevalence / intensity through improved host immunity or by supporting a greater diversity of pollinator species that are pathogen resistant (which could block transmission routes; a.k.a. a dilution effect). However, improved resources can also increase pathogen prevalence / intensity through the increased abundance of susceptible hosts, which in turn can lead to increased parasite transmission rates (from increased pollinator encounter rates), decreased immune function (from density-related stress), or increased pathogen diversity that leads to higher coinfection rates (Allander and Schmid-Hempel, 2000). Our data suggest that greater BFA and natural lands amplify infection intensity for DWV, IAPV, and LSV. Although this may be due to increased encounter rates, we speculate that increased coinfections synergistically induced higher viral loads (Durand et al., 2023), as BFA size was positively associated with colony viral richness in our study (Table 2; Fig. 3A). Previous work in bumble bees similarly showed that greater floral resource abundance was positively associated with pathogen richness among pollinator gardens, which fueled spillover between bumble bees and honey bees (Cohen et al., 2022). The BFA-pathogen richness association is interesting, as it suggests that enzootic pathogens locally fade-out as BFA declines, likely due to the local pollinator population dropping below its Critical Community Size (CCS; (Bartlett, 1960)). Under this scenario, DWV and BQCV appear to have exceeded the CCS necessary to resist fade-out, given their near ubiquitous presence in our study.

Central Florida contains numerous large and small bodies of fresh water scattered throughout the landscape (Fig. 1). The amount of fresh water surrounding a hive was also positively associated with BQCV, IAPV, and LSV infection intensities. Open water provides no floral resources for pollinators. However, if open water improves nearby resource abundance via improved hydrology or causes local pollinators to aggregate at high density on "floral islands", then OW could act like BFA and increase pollinator encounter rates and disease transmission. As noted in our results, OW was positively associated with coinfections, which could be driven by a higher

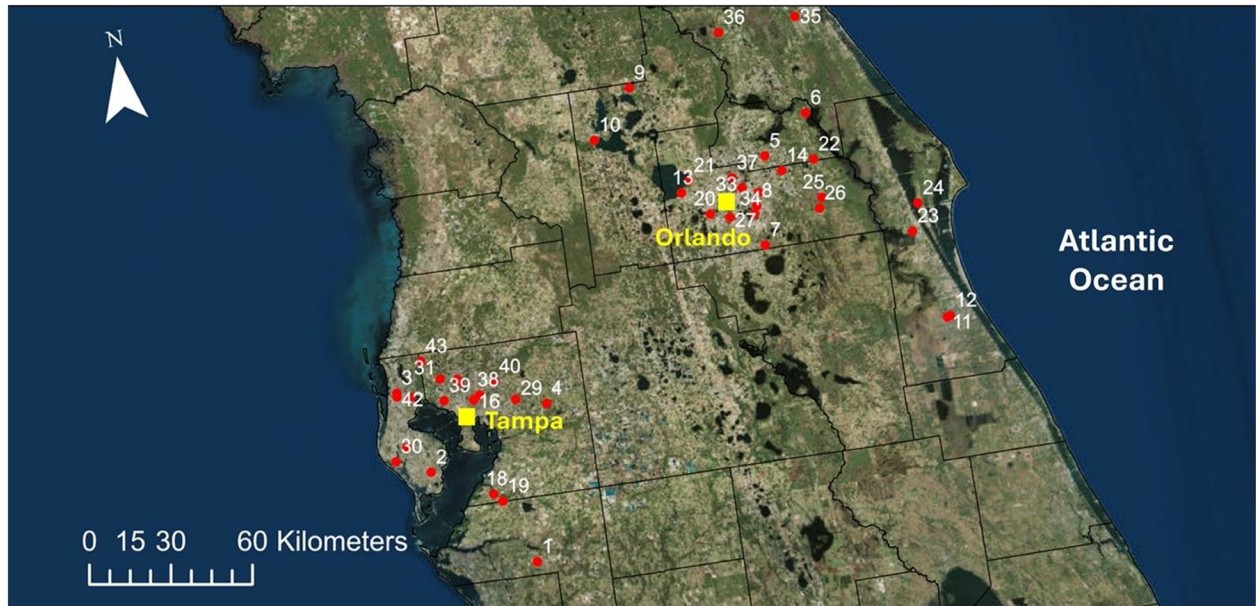

**Fig. 1. Collection site locations.** Red dots indicate colony locations around the Orlando and Tampa metropolitan areas of Central Florida. Basemap: Esri World Imagery (Clarity), © Esri, Maxar, Earthstar Geographics, CNES/Airbus DS, and the GIS User Community.

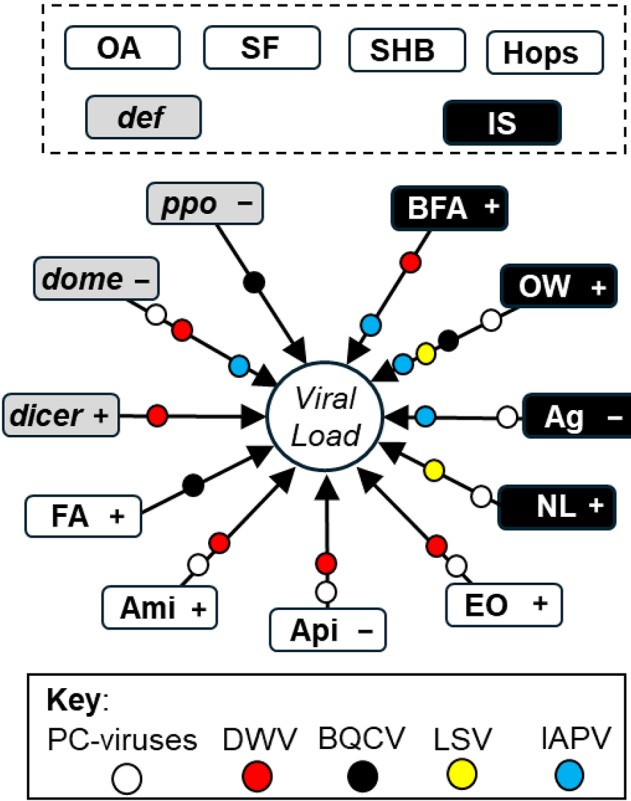

**Fig. 2. The influence of immunity, husbandry and landscape on viral infection intensity.** This figure is a consolidated representation of the statistical model results and shows the influence of landscape (black boxes), immunity (gray boxes) and husbandry (while boxes) on viral infection intensity (center circle). Colored circles represent the viral model in where an independent variable was significant. Variables within the dotted box had no detectable impact. The sign to the right of the variable depicts its direction of influence (+ or −).

pollinator population density around open water. Alternatively, open water may expose bees to pesticides commonly applied to Florida waters for mosquito and aquatic plant control, such as the herbicide glyphosate or mosquito larvicide spinosad (Brown et al., 2022.; Kaur et al., 2025). Glyphosate has been shown to alter *A. mellifera* gut microbiota and immune system regulation (Motta et al., 2018, 2022) while spinosad has been shown to alter hemocyte concentration (Kaur et al., 2025). Considering that bees utilize nearby water as hydration sources, ingestion of these water-bound pesticides would likely impair immune function and allow viral proliferation. That said, the concentration of adult mosquitos also increases near water, which leads to a greater application of mosquito adulticides, many of which have been shown to impact bee health (Zhao et al., 2022).

Regarding husbandry, beekeeping treatments are used primarily to ameliorate the effects of *V. destructor* mites, which is a primary vector of DWV, though other viruses can be transmitted (Erban et al., 2015). However, little is known about the off-target effects of these interventions on bee immunity or viral proliferation; especially those viruses not directly associated with *V. destructor*. Apiguard was the most effective treatment for reducing DWV levels in our study, with no detectable off-target effects. Surprisingly, amitraz was associated with increased DWV loads. Either this is a data/analytic artifact, or the impact of amitraz on colony health is complicated, perhaps reducing *V. destructor* numbers while promoting DWV titers through some unknown mechanism

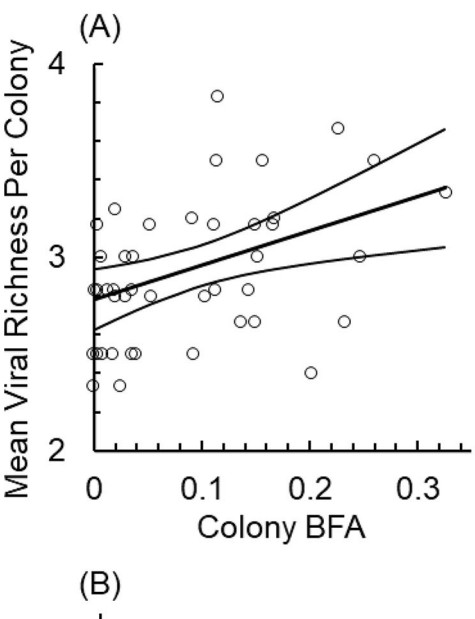

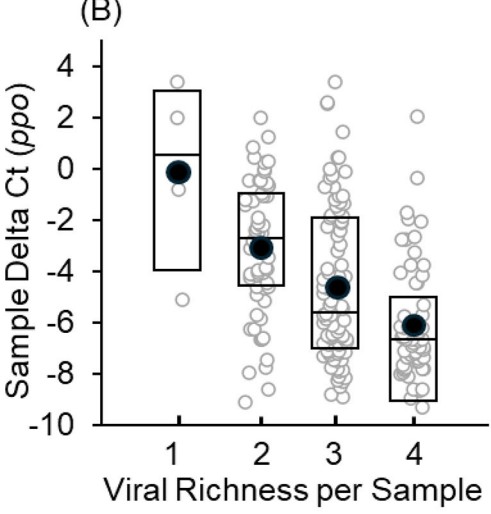

**Fig. 3. The association between coinfections, immunity, and landscape.** (A) The average number of coinfections across colonies was positively correlated with colony bee forage area (BFA). BFA estimates the proportion of a 2 km radius area surrounding a colony that contains pollination resources. (B) The association between viral richness and *ppo* ΔCt for all bee samples (r=−0.43, P<0.0001). Box plot represents median, 1st and 3rd quartile. Filled circle represents mean *ppo*.

(e.g. unrevealed immune suppression). Although mite-oriented husbandry treatments are aimed at V. destructor (and by extension DWV), they also impacted the other viruses. Formic acid was positively associated with BQCV and amitraz and essential oils did have an impact on overall viral infection intensity with PC-Virus. Thus, some husbandry treatments may be effective at decreasing *V. destructor* abundance (or DWV) as intended but influence the proliferation of other off-target pathogens. Although our husbandry

**Table 2. Impacts of landscape, immunity, and husbandry variables on average colony viral richness**

| Source | Beta | s.e. | SW | t-value | P |
|---|---|---|---|---|---|
| *domeless* | −0.26 | 0.13 | 0.86 | −2.0 | 0.0501 |
| Bee Forage Area | 0.49 | 0.13 | 1 | 3.9 | 0.0004 |
| Open Water | 0.32 | 0.13 | 1 | 2.5 | 0.0162 |
| Amitraz | 0.20 | 0.13 | 0.52 | 1.6 | 0.1236 |

analysis was coarse (i.e. simple binary predictors of use), these patterns suggest more research is needed to better understand the off-target implications of husbandry.

As expected, immune gene transcription levels tended to be negatively associated with viral loads, suggesting a robust immune system can ameliorate viral threats. This is clearly seen in the relationships between *prophenoloxidase* (*ppo*) and *domeless* (*dome*) with DWV, BQCV, and IAPV infection intensities. *Domeless* is a transmembrane receptor of the JAK-STAT pathway, while *Prophenoloxidase* is an enzyme that facilitates the conversion of tyrosine into melanin to encapsulate and dispose of invading pathogens (Lemaitre and Hoffmann, 2007). Both genes are associated with antiviral defense in honey bees (Maggi et al., 2019). However, we found a positive association between *dicer* and DWV. *Dicer* is a critical component of the RNAi pathway that prepares viral dsRNA for posttranscriptional gene suppression (Durand et al., 2023) and is active during honey bee viral infection (Galbraith et al., 2015). While the positive association is unexpected, it is important to note that co-infections with multiple viruses in honey bees can alter both viral titers and gene transcription levels in unexpected ways (Amiri et al., 2020). Specifically, *dicer* tends to be positively associated with DWV loads due to some strains evading the RNAi pathway (Durand et al., 2023). Thus, honey bees infected with DWV (which represents 98% of samples) could consequently exhibit inflated *Dicer* transcription levels.

As mentioned above, BFA represents the area of floral resources in the landscape surrounding a colony, which could be associated with improved colony resource availability. If true, increased BFA could improve colony health and immune function and reduce disease incidence (Becker and Hall, 2014). In contrast, we found that BFA was positively associated with DWV and IAPV, while its highly correlated counterpart "natural lands" was positively associated with LSV infection loads. At least two non-mutually exclusive hypotheses could account for this pattern. First, larger BFAs could offer reduced resource availability to colonies if the area maintains higher pollinator densities. This would induce host stress possibly impairing immune defenses against viral threats. Second, larger BFAs could maintain higher pathogen richness and probability of coinfection by maintaining a larger pollinator population that resists viral fade-out. This would increase a colonies' pathogen burden and reduce bee immune function. As indicated by our data immune function declines as coinfections increase, consistent with the latter hypothesis.

It is important to note that several limitations exist with our exploratory study. Relevant factors affecting bee health such as bee genetic lineage, agricultural pesticide exposure, or colony condition metrics, were not collected but could have yielded useful insights. Further, our results represent a limited geographic range (central Florida), and extrapolation to other areas should be done with caution. Larger studies at greater scales would help to clarify if these trends are ubiquitous or unique to this study's regional attributes. Last, associations for which we had no explicit predictions (e.g. BFA) leave us with *a posteriori* hypotheses that should be further researched before firm conclusions can be drawn.

In summary, our results suggest that increased floral resources and open water surrounding a colony increases viral loads, as well as pathogen richness, in managed honey bees. This is likely due to increased pollinator abundance, increased contact rates, and coinfection probability. We also show that mite-husbandry interventions can have unexpected off-target impacts on several viruses beyond DWV. Future research should work to clarify the impact of these interventions on viral loads and the mechanisms underlying the BFA, open water, immunity, and viral infection associations.

## METHODS

### Beekeeper husbandry survey, site selection and sample collection

A survey was distributed to Florida beekeepers on social media that gathered information on husbandry activities (Supplemental Document 1). The survey inquired if beekeepers used the following interventions: supplemental feeding (SF), small hive beetle management (SHB), essential oils (EO), Apiguard (Api), amitraz (Ami), hops, oxalic acid (OA), and formic acid (FA). Here, essential oils did not include thymol, the primary ingredient of Apiguard. Survey questions captured details on the quantity, frequency, and timing of each intervention; however, the responses were highly variable across participants, resulting in insufficient replication for statistical analysis. Therefore, these interventions were coded as binary variables (yes/no) for model building purposes (see statistical analyses). The survey additionally requested permission to sample colonies for viruses. In total, 94 surveys were completed and 68 respondents requested viral testing. To facilitate sampling, sites more than 100 miles from the research facility at the University of Central Florida (Orlando, FL) were excluded. Forty-three of the 68 respondents met the range limit criterion and were selected for sampling (Fig. 1). Field collections occurred in late spring 2021, which coincides with higher infection rates for several viruses (Chen et al., 2021). Collections were conducted during peak foraging hours (11:00 AM to 2:00 PM), with foragers captured at the hive entrance using a sweep net (De Miranda et al., 2013; Karbassioon and Stanley, 2023). If there was more than one hive at a site, hives were selected randomly for sample collection. At each site 24 honey bees were collected, placed in 15 ml conical centrifuge tubes, and kept alive on ice for 2±1 h until tissue could be extracted and RNA stabilized. Abdominal soft tissue—including the stinger and last integument—was extracted using the stinger-pull method described by de Miranda (2013) and homogenized in a 1.5 ml microcentrifuge tube with 400 µl of TRIzol (Invitrogen #15596026). Considering the total number of samples to be assayed was constrained, we combined 4 stinger-pulls into a single sample to improve immune gene and viral load estimates for a given colony (24 bees collected per colony to create 6 samples of 4 individuals each).

### Assessment of colony immunity and viral loads

Samples were homogenized using a motorized mortar and pestle before undergoing traditional chloroform RNA extraction per the manufacturer's instructions. RNA quantity and purity was determined using a BioTek Microplate reader (Model Synergy HTX), which provided 260:280 ratios and RNA concentration. Successful RNA extractions (260:280>1.9) were reverse transcribed into cDNA (Thermo Fisher High-Capacity cDNA RT Kit #4368813). Samples with low cDNA 260:280 ratios (<1.75) were removed to ensure quality gene transcription estimates via qPCR.

In total, four immune genes, four viruses, and one reference gene (*Actin*) were amplified for 244 samples across 43 colonies (totaling 1032 bees). DWV-A, BQCV, IAPV, and LSV-2 were chosen as they have been detected in Florida in recent surveys and are a potential concern for colony loss (Aurell et al., 2024). Viral primers were validated in-house using both synthetic gBlock and locally sourced samples that had previously tested positive through third-party diagnostics. The immune genes *prophenoloxidase* (*ppo*), *defensin* (*def*), *domeless* (*dome*), and *dicer* were examined as they represent the four major immunological pathways in insects: IMD/melanization, Toll, Jak-STAT, and RNAi, respectively (see Table S1 for primers). Although *actin* is a commonly used reference gene in honey bee viral research, it may be differentially regulated under infection conditions. This has led more recent works to use multiple reference genes in an attempt to improve accuracy, though most reference genes likely suffer from this effect (H. F. Boncristiani et al., 2013). In our study, *actin* cycle thresholds (Ct) exhibited no variation across colonies ($P=0.9998$) and multiple measures of *actin* from the same sample were highly repeatable (intraclass correlation=96.3%). Furthermore, *actin* Ct values exhibited a narrow range (19.5±0.4, mean+95% CI) and were close to the target gene/viral load Ct values supporting *actin* as an appropriate reference choice. Primer optimization informed the following thermal cycling program (Bio-Rad CFX96): 3 min at 95°C followed by 40 amplification cycles of 95°C for 15 s and 58°C for 30 s.

The magnitude of immune gene regulation was determined via relative quantitation using the ΔCt method ($\Delta Ct_{Target}=Ct_{Actin}-Ct_{Target}$; Ct=cycle

Biology Open

threshold). The magnitude of viral load was estimated as the starting copy number of the viral target in a sample, which was determined via absolute quantitation using a gBlock gene fragment of known concentration (11,261,403 copies/µl). The gBlock fragment contained sequences of Actin and the four viruses ( Table S2) and was run on each plate as a standard curve.

## Landscape characterization
Following the sampling season, apiaries and surrounding landscape variables were mapped in ArcMap 10.8.2. Three GIS datasets were used to characterize surrounding landscape including land usage (NASS-CDL), bee forage availability (FDACS-DPI), and impervious surfaces (NLCD). The high correlation between these datasets suggests accurate landscape characterization across data sources. Specifically, the FDACS-DPI variable Bee Forage Area (BFA) was strongly correlated with the Natural Lands variable in the NASS-CDL dataset (Spearman's ρ: 0.85, $P<0.001$) and negatively correlated with impervious surface from NLCD dataset (Spearman's ρ: −0.80, $P<0.001$). For each colony, landscape variables in a two-kilometer radius around the hive were estimated in square meters and converted into proportions (honey bees tend to forage no more than 2 km from the hive; (Couvillon et al., 2015). Landscape variables included the proportion of bee forage area (BFA), impervious surface (IS), natural lands (NL), agricultural lands (Ag), and open water (OW; Central Florida is replete with large and small bodies of fresh water scattered throughout the landscape).

## Statistical analyses
We used generalized linear models (GLM) to examine the association between viral copy number (a.k.a. infection intensity or viral load) and the independent landscape, husbandry, and immunity variables. It is important to note that this study is exploratory in nature and no explicit *a prior* associations were expected between specific independent variables and response variables. However, we did generally expect (1) immune gene regulation to be negatively associated with viral load indicating a robust immune response against infection, and (2) miticide husbandry interventions to be negatively associated with DWV, as miticides are generally aimed at reducing the DWV vector *V. destructor*. However, associations between landscape variables and viral load are difficult to predict. For instance, increased pollinator resources (indicated by BFA) could either amplify viral loads by increasing pollinator abundance and transmission rates or reduce viral loads by increasing the abundance of disease resistant pollinator species that block transmission routes (a.k.a. a dilution effect). To comprehensively evaluate all associations between independent and dependent variables, we employed a model selection strategy known as dredging using the R package MuMIn. Dredging in this context refers to the automated generation and comparison of all possible subsets of a full global model. Each subset model includes different combinations of predictor variables, allowing for an exhaustive search of potential influential variables based on an information-theoretic criterion such as AICc (corrected Akaike Information Criterion). This approach facilitates variable selection in complex datasets where prior knowledge is limited, while controlling for model complexity. To account for model selection uncertainty, we used MuMIn to average parameter estimates across all models within ΔAICc≤2 of the top model, weighted by their Akaike weights. This approach incorporates support from multiple plausible models rather than relying on a single best-fit model. Variable importance was quantified as the sum of Akaike weights for each predictor across these top models. Generally, if the lowest AICc model retained a marginally nonsignificant predictor (0.05<$P$-value<0.1) but exhibited a sum of Akaike weights (sw) greater than 50%, it was considered an important model predictor.

Five global models were examined, each with a different viral response variable including DWV, BQCV, IAPV, LSV, and the first principal component of the four viruses combined (PC_virus), which explained 44.7% of the variation in viral copy number. To avoid pseudoreplication, each colony variable (i.e. DWV copy number) was averaged among the colony samples, leaving 43 observations per model. All final models were assessed for multicollinearity using variance inflation factors (VIFs). All statistical analyses were conducted in R v4.3.3, using the packages lme4 and MuMIn for model dredging and selection.

## Acknowledgements
We thank the beekeepers that participated in our survey and donated bee samples for viral testing. We also wish to thank the Eastern Apicultural Society for providing funds that allowed us to complete immunological testing, and the University of Central Florida for providing Open Access funding.

## Competing interests
The authors declare no competing or financial interests,

## Author contributions
The study was conceived and designed by A.M. and K.M.F., data collection was conducted by A.M., R.W., data analysis was conducted by A.M. and K.M.F., and the manuscript was authored by A.M. and K.M.F.

## Author contributions
Conceptualization: A.M., K.M.F.; Data curation: A.M., R.W.; Formal analysis: A.M., K.M.F.; Funding acquisition: A.M.; Investigation: A.M., K.M.F.; Methodology: A.M., K.M.F., R.W.; Project administration: A.M.; Validation: A.M., K.M.F.; Visualization: A.M.; Writing – original draft: A.M., K.M.F.; Writing – review & editing: A.M., K.M.F.

## Funding
This work was partially supported by a grant from the Eastern Apicultural Society. Open Access funding provided by University of Central Florida. Deposited in PMC for immediate release.

## Data and resource availability
Data are available on Dryad (DOI: 10.5061/dryad.j6q573nt5).

## Peer review history
The peer review history is available online at https://journals.biologists.com/bio/article-lookup/doi/10.1242/bio.062201.reviewer-comments.pdf.

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
