## [Peer Review File · Biology Open]

Interactions of husbandry, landscape, and immunity in regulating viral loads for managed honey bees

Allison Malay, Kenneth Fedorka and Rachel Weavers

DOI: 10.1242/bio.062201

Editor: Kendra J. Greenlee

Review timeline

Original submission: 26 February 2025

Editorial decision: 20 May 2025

Revision received : 7 July 2025

Editorial decision: 4 August 2025

Resubmission: 7 August 2025

Accepted: 15 August 2025

Original submission

First decision letter

MS ID#: bio.061943

MS Title: Interactions of husbandry, landscape, and immunity in regulating viral loads for managed honey bees

Authors: Allison Malay, Kenneth Fedorka and Rachel Weavers

I have now reached a decision on the above manuscript.

The reviewer reports are shown at the bottom of this email or can be accessed, together with a copy of this decision letter, by going to:

As you will see, the reviewers raised a number of substantial criticisms that prevent me from accepting the paper at this stage.

They suggest, however, that a revised version might prove acceptable, if you can address their concerns. If you think that you can deal satisfactorily with the criticisms on revision, I would be pleased to see a revised manuscript. We would then return it to the reviewers.

At this stage, we also ask you to ensure your manuscript complies with our formatting guidelines. Provided you are able to fully address the referees' comments, we are positive about publication of your paper (we accept over 95% of revision submissions) and therefore hope you won't mind any extra work involved in reformatting your manuscript at this point.

Please ensure that you clearly highlight all changes made in the revised manuscript. Please avoid using 'Tracked changes' in Word files as these are lost in PDF conversion.

I should be grateful if you would also provide a point-by-point response detailing how you have dealt with the points raised by the reviewers in the 'Response to Reviewers' box. Please attend to all of the reviewers' comments. If you do not agree with any of their criticisms or suggestions please explain clearly why this is so.

Reviewer 1:

1. Line 28 (also lines 61-63; 295): Right off the bat, the opening sentence in the abstract repeats a common misconception that honey bees are experiencing a "decline," which is not true. There are widespread, unsustainable die-offs of managed honey bee colonies, but beekeepers are able to grow the population back from the survivors, so the population is oscillating but not declining. Solitary native bee populations, however, are likely declining but we don't have great baseline data on them so it's harder to tell. Understanding and communicating this nuance is critical for the scientific literature even if the media is constantly confounding these facts.
RESPONSE: We have revised the manuscript to more accurately characterize the current status of managed honey bee populations by emphasizing high mortality rates rather than population decline. Specifically, we revised line 63 to highlight the high annual losses rather than implying a global population decline.
2. Line 83: The "hive" is the structure in which the bees live, but the "colony" is the living superorganism. Thus "hives" are not impacted by the surrounding environment, but the colonies. This is a recurring misuse of semantics throughout the manuscript (line 90, 93, 98, etc...).
RESPONSE: We replaced references to "hives" with "colonies" where appropriate to more accurately reflect the biological entity being discussed. These changes have been made throughout the text, including at lines 78,81,93, and in other relevant instances.
3. Line 94: While Varroa was introduced and cited earlier in the introduction, the small hive beetle was not.
RESPONSE: We have added a brief introduction and appropriate citation for the small hive beetle earlier in the manuscript (lines 68-70).
4. Line 95-97: the most common synthetic acaricide is amitraz not thymol, and I think it would help to cite any one of the several reviews on Varroa IPM here.
RESPONSE: We have revised the text to identify amitraz as the example acaricide. In addition, we have added a citation to a recent review on *Varroa* integrated pest management (line 98).
5. Line 116: Surveys of beekeepers are fine, but using their results as scientific data can be dubious. Beekeepers are not always forthcoming in their management practices, and there did not seem to be any ground-truthing. The questions are fairly simple and treated as binary for the analysis, but this is missing a ton of nuance and other information. For example, supplemental feeding or not (SF) misses completely how much, how often, and what macronutrient are involved. Six of the eight management variables are also germane to a single factor (Varroa control), and again misses the details of how these were applied. It would have been far more useful, and I think biologically more appropriate, to have Varroa loads measured at the time of sampling (although I completely recognize the logistical difficulty in doing so). There are also a near infinite number of other factors that could important here (e.g., colony size, frames of brood, level of shading of the hive, if the colony overwintered or started from a newly established package), and arguably perhaps more important than what was surveyed, so it is unfortunate that the eight management variables were fairly limited in their utility and scope compared to the alternatives.
RESPONSE: We agree that survey data have inherent limitations, including variability in beekeeper reporting and a lack of detail on the specific implementation of management practices. We have added a statement to clarify that although additional questions were included in the survey to capture more nuance (e.g., frequency, quantity, or method of application), high variability and insufficient data quality led to the exclusion of those variables from the analysis (lines 124-127). While we recognize that direct measures such as *Varroa* loads and additional environmental and colony-level factors (e.g., brood size, overwintering status) would be ideal, these data were beyond the scope and logistics of this

exploratory study. Nonetheless, we believe the management variables included still offer a coarse but useful lens into beekeeper practices that may influence colony health, and serve as a foundation for identifying patterns worthy of deeper investigation in future work.

6. Line 128: Provide a citation that demonstrates that peak viral "season" is in the late spring, because I am not convinced of that based on much of the literature with which I am familiar.

RESPONSE: We have revised the language to state that this period "coincides with higher infection rates for several viruses" and added a supporting citation (Chen et al., 2021) at lines 133-134 to clarify and substantiate this point.

7. Line 129: Bees were collected with a sweep net?! This is not standard practice (see de Miranda et al. 2013, cited by the authors) and can potentially and significantly skew the results. Older workers have higher virus titers and different immunity from younger workers, but heavily infected workers may not live to flight age. Of course all this is all relative to within this study, but it makes it much harder to compare and interpret among other studies in this paradigm.

RESPONSE: To collect bees, we followed the protocol outlined in section 4.3.1 of de Miranda et al. (2013), which identifies collection at the hive entrance as a standard method for sampling foragers, similar to other relevant honey bee research (e.g., Evans et al., 2013; Chen et al., 2006). The only minor difference was the use of a sweep net instead of a brush to transfer bees into their receptacle for cold storage. This method was applied consistently across all sampling sites in our study to ensure comparability.

8. Line 131: The bees were chilled on ice but still alive, correct? RNA extraction from narcotic tissue for these RNA viruses is very dubious and variable, even after 2 hours.

RESPONSE: We have clarified in the manuscript (line 138) that bees were kept alive on ice until RNA extractions were performed, which occurred within a 2-hour window from collection to extraction to ensure RNA integrity.

9. Lines 132-137: So each colony is represented by a pool of four individual forager guts? Again, not at all conventional practice (although not necessarily wrong) but opens up the opportunity for increased variability in the data.

RESPONSE: Each colony is represented by six samples, with each sample consisting of four individual forager guts. While pooling is less conventional than analyzing individual bees, this approach was chosen due to financial/logistical constraints, and that it reduced (not increased) immunological and viral load variation within colonies compared with 6 samples comprised of 1 individual each.

10. Line 147: It's becoming more standard practice to use multiple control genes. Moreover, Actin has been shown to be very unreliable since it can be differentially regulated as a result of infection (Boncristiani et al. 2013, not cited).

RESPONSE: We recognize that the use of multiple reference genes is increasingly recommended to ensure accurate normalization in qPCR analyses. While our study used Actin as a single reference gene, we chose it based on prior validation in similar contexts. We have now cited Boncristiani et al. (2013) to acknowledge the potential limitations of Actin under infection conditions (lines 162-169).

Importantly, actin showed no variation across colonies ($P=0.9998$) and multiple measures of actin from the same sample were highly repeatable (96.3%; intraclass correlation).

Furthermore, average actin CT values were 19.5 ± 0.4 (95% CI), exhibiting a narrow range close to the CT values of the target genes (~ 24). We include these details in the revision.

11. Line 148: So there were 6 samples per colony? In that there were 6 separate pools of 4 forager guts? So really the sample size here is 43 colonies across 43 beekeeping operations, so I'm assuming one hive each? Also, the narrative thus far about the viruses measured imply that they are singularly unique, which is not true. DWV and LSV are well known to have multiple variants that vary in their virulence. I'm forgetting the details in the cited references in the primer supplemental table whether or not they are universal primers or specific to a given variant, but there should be more attention paid to these nuances in the pathogen screening protocols.

RESPONSE: We confirm that each colony is represented by six separate samples, each a pool of four forager guts, resulting in a total of 43 colonies sampled across 43 beekeeping

operations, with one colony per operation. Regarding viral variants, we acknowledge that viruses such as DWV and LSV include multiple variants with differing virulence profiles. In this study, two of the primers are universal (BQCV and IAPV), while two are variant-specific (DWV-A and LSV2), as specified at lines 155-156.

12. Line 151: How were the samples known to be infected? Because they had tested positive before? Why not use a standard curve of known concentrations? (I see the gBlock fragment was used in line 163)

RESPONSE: We have clarified in the manuscript that viral primers were validated in-house using both synthetic gBlock fragments and locally sourced samples that had previously tested positive through third-party diagnostic screening (lines 157-159). This dual approach ensured that our primers amplified both synthetic targets and biologically relevant viral sequences.

13. Line 161: While I find it appropriate to use the single control gene to calculate the delta Cts of the immune genes, it is not appropriate to use this approach for the viral targets. This is because the immune genes and actin are both honey bee genes expressed by the honey bee host, but the viral genes are not. It is better to just use the absolute data for the pathogen incidence and intensity.

RESPONSE: As suggested, the data were analyzed using the Actin-based delta CTs for the immune genes and the gBlock-based absolute data for the pathogen intensity. The results are consistent with the original submission. The new approach is reported in the revised manuscript (lines 172-175).

14. Line 196: I think the path analysis is appropriate here and well described, as I am familiar with it but not intimately so the narrative in this section is welcomed. Similarly, the caveat in line 220 is helpful and important, as the experimental design is less of a direct test of a controlled hypothesis but rather searching for patterns. This is an important initial step in epidemiological study. All of that said, I am very concerned that there was no mention of how the repeated measures per colony was taken into account (6 samples of 4 foragers per colony constitutes pseud-replication if they are not included as repeated measures) or the lack of replication of colonies per beekeeper/location (43 samples across 43 beekeepers). The variation within operations (that is, variation among colonies of the same beekeeper) is tremendous, so I would imagine that any landscape affects would be immediately swamped by intra-apiary variation. I could be misunderstanding the sampling regime here but further clarity is definitely needed. All of the results and discussion is contingent on this, so I will not comment on those sections.

RESPONSE: The data were reanalyzed using colony as a random factor. The new results failed to resolve indirect path effects (e.g. landscape effects on immunity or husbandry effects on immunity). However, the direct path effects were all similar, leading to the same overall conclusions. Given the lack of indirect effects, we switch analyses and now use GLMMs, as readers are more familiar with this type of analysis.

Reviewer 2:

1. In the introduction, the authors state that DWV virulence is, in part, linked to its mite vector *Varroa*. There is some nuance here, DWV is spread in many insects in lots of different ways, and *Varroa* is just one of them. I would state that *Varroa* is one key method of infection, but not the only one.

RESPONSE: We agree that *Varroa* is a major but not exclusive vector of DWV, and that the virus can be transmitted through multiple routes and among various insect hosts. We have revised the sentence in the Introduction to reflect this nuance by describing *Varroa* as a key, but not sole, mode of transmission (lines 73-77).

2. The tissue used for the gene expression and virus load is a little odd. The authors use the tissue that comes out when the sting is pulled. This is a little inexact. In my hands, when you pull the stinger of a bee, you can get venom gland, the Malpighian tubules, muscles, oviduct and ovary, or just some of those, or complex mixtures of them. How did the authors ensure that they had consistent samples, particularly given that the viruses they are looking

for may differ in amounts in various tissues, and gene expression certainly does. It would be good to see the authors explain what tissues they are using.

RESPONSE: We appreciate the reviewer's comment. We agree that the stinger-pull method can result in a mixture of abdominal tissues, and that this has implications for both viral load and gene expression. However, this method is well established in honey bee viral research and is described as a standard protocol (section 4.3.5 in de Miranda et al., 2013), which we have cited in the manuscript. We made every effort to minimize variation in tissue allotment among stinger pulls, although some variation likely exists. Importantly, this variation would be randomly distributed within (4 stinger pulls per sample) and among samples (representing 1032 stinger pulls). If this variation is large enough, it would reduce our capacity to detect statistical signals by contributing error variance to our statistical models. That we found strong statistical signals suggests that this variation was minimal.

3. IN the qPCR, the authors do two things. a) They do relative gene expression analysis, and they do absolute quantitation. For the relative quantitation, the authors use actin as their reference (note NOT housekeeping) gene. For these sorts of analyses, it is standard practice to use two to three reference genes and use the geometric mean of those references as the standard they measure against. Why was this not done here? Also, I don't see data on the ct values in the manuscript or the supplemental information. Without this information, it is hard to judge if actin is an appropriate reference gene- does have ct values in the range that the virus and immune genes do? This data should be included, and if actin doesn't provide a suitable reference, then new references should be found. The authors also do not show whether actin is stable between samples in this analysis. IT should be, but given the possibility of variation in the testing tissue (see 2 above) it would be worth ensuring it is stable. This manuscript reports important things, it would be a pity for that to be undermined by a poor reference gene.

RESPONSE: We appreciate the reviewer's detailed feedback and agree that reference gene validation is essential for reliable relative gene expression analysis. Actin was used as the reference gene based on its common use in honey bee studies and prior reports of stability in similar contexts. In response to this comment, we have updated the manuscript to consistently refer to Actin as a reference gene rather than a "housekeeping gene" to reflect proper terminology (line 154). Additionally, we have made available the full data set which contains the Ct values for Actin and all target genes. Average ct values \pm 95% CIs for *actin*, *defensin*, *dicer*, *dome*, *ppo*, DWV, BQCV, LSV, and IAPV were 19.5 ± 0.4 , 22.2 ± 0.5 , 25.1 ± 0.4 , 23.6 ± 0.26 , 23.9 ± 0.21 , 24.4 ± 1.0 , 26.7 ± 0.7 , 33.0 ± 1.2 , 34.7 ± 0.9 , respectively. These were well within detection ranges of qPCR and exhibited stability. Furthermore, actin showed relatively low variation across colonies ($P=0.9998$) and multiple measures of actin from the same sample were highly repeatable (96.3%; intraclass correlation). We include these actin details in the revision. We acknowledge that the inclusion of multiple reference genes would improve the robustness of relative expression estimates, and have also added this point as a limitation in lines 162-169.

4. IN the results section, we meet a whole load of abbreviations that are not introduced. The authors need to make sure these are defined earlier- or, in my preference, avoid them. The results would be a much easier read if I didn't have to keep trying to find what each abbreviation means.

RESPONSE: Thank you for this helpful suggestion. We have reviewed the manuscript and ensured that all abbreviations are defined upon first use in either the Introduction or Methods sections. Where possible, we have reduced the use of abbreviations to improve readability, particularly in the Results section, in line with the reviewer's preference.

5. Place keep to standard nomenclature for genes, etc. For example, *dicer* should be lowercase italics without a capital if talking about the gene.

RESPONSE: We have reviewed the manuscript and updated gene names to follow standard nomenclature conventions, including italicizing gene symbols and using lowercase letters where appropriate (e.g., *dicer*).

6. Line 336 states Apigard has no off-target effects- should be no off-target effects detected in this study.

RESPONSE: We have revised the sentence to state that “no off-target effects were detected in this study” to more accurately reflect the scope of our findings (line 344).

7. Figure 1 is a terrible map- please put a good one in that is at least informative
RESPONSE: We have replaced Figure 1 with a revised and more informative map that includes clearer labeling, improved resolution, and relevant geographic context to better support the manuscript’s content.
8. Figure 2 needs much more explanation- please provide a figure legend that explains it.
RESPONSE: This figure was removed from the revision.

Author response to reviewers’ comments

Reviewer 1: General comments

This study represents a huge undertaking, looking for patterns of honey bee health with respect to landscape variables, beekeeper management decisions, and immunity measures. At issue is the well-documented honey bee colony losses and the threat they pose on pollination services in commercial production agriculture. This is a very noisy, highly variable system in which to study these interactions, and the experimental design is ambitious. My sense is that there are very important and insightful results in this dataset, but there are some inconsistencies in concept and execution that cast doubt in my mind about the currently drawn inferences. There are some critically important details that are either wrong or glazed over that could significantly affect the analysis and results, so I think a more careful treatment of them would improve the strength of the report.

Specific comments

1. Line 28 (also lines 61-63; 295): Right off the bat, the opening sentence in the abstract repeats a common misconception that honey bees are experiencing a “decline,” which is not true. There are widespread, unsustainable die-offs of managed honey bee colonies, but beekeepers are able to grown the population back from the survivors, so the population is oscillating but not declining. Solitary native bee populations, however, are likely declining but we don’t have great baseline data on them so it’s harder to tell. Understanding and communicating this nuance is critical for the scientific literature even if the media is constantly confounding these facts.
2. Line 83: The “hive” is the structure in which the bees live, but the “colony” is the living superorganism. Thus “hives” are not impacted by the surrounding environment, but the colonies. This is a recurring misuse of semantics throughout the manuscript (line 90, 93, 98, etc...).
3. Line 94: While Varroa was introduced and cited earlier in the introduction, the small hive beetle was not.
4. Line 95-97: the most common synthetic acaricide is amitraz not thymol, and I think it would help to cite any one of the several reviews on Varroa IPM here.
5. Line 116: Surveys of beekeepers are fine, but using their results as scientific data can be dubious. Beekeepers are not always forthcoming in their management practices, and there did not seem to be any ground-truthing. The questions are fairly simple and treated as binary for the analysis, but this is missing a ton of nuance and other information. For example, supplemental feeding or not (SF) misses completely how much, how often, and what macronutrient are involved. Six of the eight management variables are also germane to a single factor (Varroa control), and again misses the details of how these were applied. It would have been far more useful, and I think biologically more appropriate, to have Varroa loads measured at the time of sampling (although I completely recognize the logistical difficulty in doing so). There are also a near infinite number of other factors that could important here (e.g., colony size, frames of brood, level of shading of the hive, if the colony overwintered or started from a newly established package), and arguably perhaps more important than what was surveyed, so it is unfortunate that the eight management variables were fairly limited in their utility and scope compared to the alternatives.

6. Line 128: Provide a citation that demonstrates that peak viral "season" is in the late spring, because I am not convinced of that based on much of the literature with which I am familiar.
7. Line 129: Bees were collected with a sweep net?! This is not standard practice (see de Miranda et al. 2013, cited by the authors) and can potentially and significantly skew the results. Older workers have higher virus titers and different immunity from younger workers, but heavily infected workers may not live to flight age. Of course all this is all relative to within this study, but it makes it much harder to compare and interpret among other studies in this paradigm.
8. Line 131: The bees were chilled on ice but still alive, correct? RNA extraction from narcotic tissue for these RNA viruses is very dubious and variable, even after 2 hours.
9. Lines 132-137: So each colony is represented by a pool of four individual forager guts? Again, not at all conventional practice (although not necessarily wrong) but opens up the opportunity for increased variability in the data.
10. Line 147: It's becoming more standard practice to use multiple control genes. Moreover, Actin has been shown to be very unreliable since it can be differentially regulated as a result of infection (Boncristiani et al. 2013, not cited).
11. Line 148: So there were 6 samples per colony? In that there were 6 separate pools of 4 forager guts? So really the sample size here is 43 colonies across 43 beekeeping operations, so I'm assuming one hive each? Also, the narrative thus far about the viruses measured imply that they are singularly unique, which is not true. DWV and LSV are well known to have multiple variants that vary in their virulence. I'm forgetting the details in the cited references in the primer supplemental table whether or not they are universal primers or specific to a given variant, but there should be more attention paid to these nuances in the pathogen screening protocols.
12. Line 151: How were the samples known to be infected? Because they had tested positive before? Why not use a standard curve of known concentrations? (I see the gBlock fragment was used in line 163)
13. Line 161: While I find it appropriate to use the single control gene to calculate the delta Cts of the immune genes, it is not appropriate to use this approach for the viral targets. This is because the immune genes and actin are both honey bee genes expressed by the honey bee host, but the viral genes are not. It is better to just use the absolute data for the pathogen incidence and intensity.
14. Line 196: I think the path analysis is appropriate here and well described, as I am familiar with it but not intimately so the narrative in this section is welcomed. Similarly, the caveat in line 220 is helpful and important, as the experimental design is less of a direct test of a controlled hypothesis but rather searching for patterns. This is an important initial step in epidemiological study. All of that said, I am very concerned that there was no mention of how the repeated measures per colony was taken into account (6 samples of 4 foragers per colony constitutes pseud-replication if they are not included as repeated measures) or the lack of replication of colonies per beekeeper/location (43 samples across 43 beekeepers). The variation within operations (that is, variation among colonies of the same beekeeper) is tremendous, so I would imagine that any landscape affects would be immediately swamped by intra-apiary variation. I could be misunderstanding the sampling regime here but further clarity is definitely needed. All of the results and discussion is contingent on this, so I will not comment on those sections.

Reviewer 2: Malay et al present an interesting manuscript that aims to understand the relationships between various hive and environmental factors and viral loads in bee hives. This is interesting as we need as much data as possible to understand how best to support beehives, and much of the current literature is not rigorous due to the difficulty of these experiments.

The authors present a compelling set of measurements and statistical approaches that help them link environmental and gene expression factors with pathogen loads. This approach is useful and gives interesting and unexpected findings, some of which have real consequences for hive management.

The manuscript is clear and well written, but I have some suggestions that I think should be addressed,, some of which are minor issues.

- 1) In the introduction, the authors state that DWV virulence is, in part, linked to its mite vector Varroa. There is some nuance here, DWV is spread in many insects in lots of different ways, and Varroa is just one of them. I would state that Varroa is one key method of infection, but not the only one.
- 2) The tissue used for the gene expression and virus load is a little odd. The authors use the tissue that comes out when the sting is pulled. This is a little inexact. In my hands, when you pull the stinger of a bee, you can get venom gland, the Malpighian tubules, muscles, oviduct and ovary, or just some of those, or complex mixtures of them. How did the authors ensure that they had consistent samples, particularly given that the viruses they are looking for may differ in amounts in various tissues, and gene expression certainly does. It would be good to see the authors explain what tissues they are using.
- 3) IN the qPCR, the authors do two things. a) They do relative gene expression analysis, and they do absolute quantitation. For the relative quantitation, the authors use actin as their reference (note NOT housekeeping) gene. For these sorts of analyses, it is standard practice to use two to three reference genes and use the geometric mean of those references as the standard they measure against. Why was this not done here? Also, I don't see data on the ct values in the manuscript or the supplemental information. Without this information, it is hard to judge if actin is an appropriate reference gene- does have ct values in the range that the virus and immune genes do? This data should be included, and if actin doesn't provide a suitable reference, then new references should be found. The authors also do not show whether actin is stable between samples in this analysis. IT should be, but given the possibility of variation in the testing tissue (see 2 above) it would be worth ensuring it is stable. This manuscript reports important things, it would be a pity for that to be undermined by a poor reference gene.
- 4) IN the results section, we meet a whole load of abbreviations that are not introduced. The authors need to make sure these are defined earlier- or, in my preference, avoid them. The results would be a much easier read if I didn't have to keep trying to find what each appreciation means.
- 5) Place keep to standard nomenclature for genes, etc. For example, dicer should be lowercase italics without a capital if talking about the gene.
- 6) Line 336 states Apigard has no off-target effects- should be no off-target effects detected in this study.
- 7) Figure 1 is a terrible map- please put a good one in that is at least informative
- 8) Figure 2 needs much more explanation- please provide a figure legend that explains it.

In terms of the review rubric provided...

Each figure has a proper control- though it is important to be clear whether the control gene in the relative qPCR analysis is the right one, and explain why the authors are not using the standard approach to quantisation (i.e. using the geometric mean of a few reference genes)

Are experiments performed using appropriate methods that will answer the question (or test the hypothesis or support the observations) posed by the authors? Is the right tool used for the job? I think the approach, apart for the qPCR which needs to be explained, is a good one and appropriate.

Were the data analyzed using appropriate statistical tests? Yes

Reproducibility

Were experiments in each figure performed using adequate number of biological replicates? Yes- and this is hard in beekeeping research!

Is there sufficient raw data to assess the rigor of the analysis? I think it would be good to see some of the raw data, especially the quality controls for the qPCR in the manuscript as supplemental material. I note the raw data is ti be deposited in an open access archive.

Does the methods section provide sufficient detail to permit reproducibility? Yes

Completeness

Are the author's conclusions supported by the data? Yes

Are there any flaws in the experimental design that invalidate the approach taken by the authors?
No, but the qPCR needs explaining

Are there experiments that have not been performed, but if true would disprove the conclusion? If yes, and if such experiments would be costly or time-consuming to perform, do the authors acknowledge this in a discussion of the limitations? The authors are very clear as to the limitations of their experiments.

Scholarship

Do the authors cite and discuss the merits of relevant data that would argue against their conclusion? Yes

Do the authors cite and discuss the merits of relevant data that would support their conclusion? Yes

Reviewer's Responses to Questions

Experimental quality

Does each figure have the proper controls?

If 'No', please indicate reasons in Comments for Author box below.

Reviewer #1:

- No

Reviewer #2:

- Yes

Were the data analyzed using appropriate statistical tests?

If 'No', please indicate reasons in Comments for Author box below.

Reviewer #1:

- Yes

Reviewer #2:

- Yes

Reproducibility

Were experiments performed using adequate number of biological replicates?

If 'No', please indicate reasons in Comments for Author box below.

Reviewer #1:

- No

Reviewer #2:

- Yes

Does the methods section provide sufficient detail to permit reproducibility?

If 'No', please indicate reasons in Comments for Author box below.

Reviewer #1:

- No

Reviewer #2:

- Yes

Completeness

Are the manuscript's conclusions supported by the data?

If 'No', please indicate reasons in Comments for Author box below.

Reviewer #1:

- No

Reviewer #2:

- Yes

Scholarship

Do the authors cite and discuss the merits of data that would argue for and against their conclusion?

If 'No', please indicate reasons in Comments for Author box below.

Reviewer #1:

- Yes

Reviewer #2:

- Yes

Does the manuscript title & abstract accurately reflect the contents of the manuscript, without hyperbole?

If 'No', please indicate reasons in Comments for Author box below.

Reviewer #1:

- Yes

Reviewer #2:

- Yes

Resubmission

First decision letter

MS ID#: bio.062201

MS Title: Interactions of husbandry, landscape, and immunity in regulating viral loads for managed honey bees

Authors: Allison Malay, Kenneth Fedorka and Rachel Weavers

I have now reached a decision on the above manuscript.

The reviewer reports are shown at the bottom of this email or can be accessed, together with a copy of this decision letter, by going to:

Reviewer 1 raised an important concern regarding the sampling and statistical analysis that is significant enough to prevent me from accepting the paper for publication. I am sorry to write with this disappointing news

Having said that, should you be able to adequately address the reviewer's concern regarding the sampling, then I would be happy to see the paper again, as a new submission. The reviewer is concerned that the samples from each colony makes that a repeated measure. The sample unit should be colony, which makes the sample size 43 rather than 244. The reviewer suggests that the samples should be re-run with the individual stingers; however it seemed from the methods that the samples were pooled at the collection time, which would make that impossible. At a minimum, the repeated nature of the sampling should be taken into account during the statistical analyses. If after considering the feedback, you instead decide to submit elsewhere, please let me know, so that we can close our file.

Comments from the Reviewers:

Reviewer 1: The revised submission greatly improves the readability and accuracy of the manuscript, as well as providing clarity on the experimental design. However, through that clarity, it seems to me that the experimental design was just inappropriate, or perhaps misapplied, for the study questions. It makes no sense (or I should say, there is zero benefit) to have 6 separate samples of 4 bees for each colony and only one colony for each beekeeper (unless one wants to capture the intracolony variability, which was not the point of the study). At the very least all 24 bees should have been pooled for each colony (which actually might be possible with the samples in the freezer), but it seems like a missed opportunity to survey multiple colonies per operation using a repeated measures design to increase the statistical power for the management practices and other variables. The author response about this being an "exploratory study" as the reason more key data were not taken (especially Varroa mite infestation levels) was particularly disappointing since the study conclusions draw inferences that are much more than descriptive. In the end, with colony as the basal unit, the sample size is roughly that of the variables measured, so the noise of the independent variables (especially the generic survey answers) likely swamps any true signal with which we can have much confidence.

Reviewer 2: The authors have addressed my previous concerns with their manuscript well. My key concerns were the sampling used for the gene expression analysis, which, while not ideal, I think is now better explained and provides me more confidence that the results are robust. I was also concerned with the qPCR reference used, as the authors only used one reference gene and provided no information about its stability or relationship to the other genes measured. The authors have provided that data, and I am confident that actin is providing a good reference in this case. Please, however, for any future analyses of this sort, use multiple reference genes. I am convinced that poor reference genes are responsible for some of the more outlandish 'findings' in the honeybee literature, and it would be good to ensure that all such analyses are done the best way possible. In this case, you are lucky that your reference gene is ok. Can you also check the correct name of your reference gene? You name it as Actin 5C, which is the name of a Drosophila actin gene. Is the actin copy you used the most similar to Actin5C, or is it named this in the latest annotation of the genome. Annotations in bees are all a bit messy- would it be possible to add a table that links the gene names you have to LOC numbers for these genes- I think this would remove any confusion and help people navigate any more revisions of the honeybee genome.

Reviewer's Responses to Questions

Experimental quality

Does each figure have the proper controls?

If 'No', please indicate reasons in Comments for Author box below.

Reviewer #1:

- Yes

Reviewer #2:

- Yes

Were the data analyzed using appropriate statistical tests?

If 'No', please indicate reasons in Comments for Author box below.

Reviewer #1:

- Yes

Reviewer #2:

- Yes

Reproducibility

Were experiments performed using adequate number of biological replicates?

If 'No', please indicate reasons in Comments for Author box below.

Reviewer #1:

- No

Reviewer #2:

- Yes

Does the methods section provide sufficient detail to permit reproducibility?

If 'No', please indicate reasons in Comments for Author box below.

Reviewer #1:

- Yes

Reviewer #2:

- Yes

Completeness

Are the manuscript's conclusions supported by the data?

If 'No', please indicate reasons in Comments for Author box below.

Reviewer #1:

- No

Reviewer #2:

- Yes

Scholarship

Do the authors cite and discuss the merits of data that would argue for and against their conclusion?

If 'No', please indicate reasons in Comments for Author box below.

Reviewer #1:

- Yes

Reviewer #2:

- Yes

Does the manuscript title & abstract accurately reflect the contents of the manuscript, without hyperbole?

If 'No', please indicate reasons in Comments for Author box below.

Reviewer #1:

- No

Reviewer #2:

- Yes

Author response to reviewers' comments

Dear Dr. Kendra Greenlee

Thank you for allowing us to resubmit. We have reanalyzed the data as suggested by reviewer #1 and yourself, and have collapsed our data structure to the level of the colony. As you can see from our revision, the results are largely the same and our conclusions remain unchanged. As we note in our response below, such robustness suggests that the patterns we reported in our earlier submission were not an artifact of random-effect models or pseudoreplication but reflect genuine associations. As we feel we have adequately addressed the statistical issues raised, we ask that our current revised submission will be reevaluated for publication.

Reviewer 1:

1. It makes no sense (or I should say, there is zero benefit) to have 6 separate samples of 4 bees for each colony and only one colony for each beekeeper (unless one wants to capture the intracolony variability, which was not the point of the study). At the very least all 24 bees should have been pooled for each colony (which actually might be possible with the samples in the freezer), but it seems like a missed opportunity to survey multiple colonies per operation using a repeated measures design to increase the statistical power for the management practices and other variables. In the end, with colony as the basal unit, the sample size is roughly that of the variables measured, so the noise of the independent variables (especially the generic survey answers) likely swamps any true signal with which we can have much confidence.

RESPONSE: We agree with the reviewers' point that the primary focus of the manuscript was to assess the immune system, landscape, and husbandry associations with colony viral load. From this perspective, colony would be the most appropriate unit of analysis (as the editor also points out). Thus, we reanalyzed the data by averaging sample metrics at the colony level and avoid any possibility of pseudoreplication. Importantly, the results of these analyses are highly consistent with our original findings using random effect models, and our conclusions remain unchanged. This robustness suggests that the patterns we report were not an artifact of random-effect models or pseudoreplication, but reflect genuine associations. We have revised our manuscript accordingly with the new analysis using colony as the experimental unit.

A minor focus of the manuscript was to assess associations between immunity and viral richness (excluding landscape and husbandry). As opposed to the "colony-level" question above, this can be addressed at the individual level, as individuals (or samples) vary in their viral richness, immune responses, and viral intensities. This is a minor aspect of the paper (only Figure 3B) and can be analyzed as a random effects model using colony as the random factor without concern of pseudoreplication. Random effects models are a more flexible and powerful way to handle repeated measures data and used all 244 samples. As such, it provides a more robust assessment of how viral richness and immunity are related.

Reviewer 2: The authors have addressed my previous concerns with their manuscript well. Can you also check the correct name of your reference gene? You name it as Actin 5C, which is the name of a *Drosophila* actin gene. Is the actin copy you used the most similar to Actin5C, or is it named this in the latest annotation of the genome. Annotations in bees are all a bit messy- would it be possible to add a table that links the gene names you have to LOC numbers for these genes- I think this would remove any confusion and help people navigate any more revisions of the honeybee genome.

RESPONSE: We thank the reviewer for noting that “Actin 5C” is a *Drosophila* gene name. Its use in our manuscript was an inadvertent carryover from *Drosophila* literature. We have updated the manuscript to reflect the correct name. We also updated Supplemental Table 1 with the citations from where the primers were derived and the ascension numbers.

Second decision letter

MS ID#: bio.062201R1

MS Title: Interactions of husbandry, landscape, and immunity in regulating viral loads for managed honey bees

Authors: Allison Malay, Kenneth Fedorka and Rachel Weavers

I appreciate the additional statistical analysis and consequent updates to the results. I am happy to tell you that your manuscript has been accepted for publication in *Biology Open*, pending our standard publication integrity checks. It was accepted on 15th August 2025.